# GAN-BASED GAUSSIAN MIXTURE MODEL RESPONSIBILITY LEARNING

## ABSTRACT

Mixture Model (MM) is a probabilistic framework which allows us to define a dataset containing $K$ different modes. When each of the modes is associated with a Gaussian distribution, we refer it as Gaussian MM, or GMM. Given a data point $x$, GMM may assume the existence of a random index $k \in \{1, \ldots, K\}$ identifying which Gaussian the particular data is associated with. In a traditional GMM paradigm, it is straightforward to compute in closed-form, the conditional likelihood $p(x|k, \theta)$, as well as responsibility probability $p(k|x, \theta)$ which describes the distribution index corresponds to the data. Computing the responsibility allows us to retrieve many important statistics of the overall dataset, including the weights of each of the modes. Modern large datasets often contain multiple unlabelled modes, such as paintings dataset containing several styles; fashion images containing several unlabelled categories. In its raw representation, the Euclidean distances between the data do not allow them to form mixtures naturally, nor it's feasible to compute responsibility distribution, making GMM unable to apply. To this paper, we utilize the Generative Adversarial Network (GAN) framework to achieve an alternative plausible method to compute these probabilities at the data's latent space $z$ instead of $x$. Instead of defining $p(x|k, \theta)$ explicitly, we devised a modified GAN to allow us to define the distribution using $p(z|k, \theta)$, where $z$ is the corresponding latent representation of $x$, as well as $p(k|x, \theta)$ through an additional classification network which is trained with the GAN in an "end-to-end" fashion. These techniques allow us to discover interesting properties of an unsupervised dataset, including dataset segments as well as generating new "out-distribution" data by smooth linear interpolation across any combinations of the modes in a completely unsupervised manner.

## 1 INTRODUCTION

Gaussian Mixture Model (GMM) is one of the commonly used probabilistic models for dataset enjoy multiple modes. It assumes that all data points come from a mixture of a finite number of Gaussian distributions. The density function of the GMM is defined below:

$$p_{\mathcal{Z}}(\mathbf{z}) = \sum_{k=1}^{K} \alpha_k \mathcal{N}(\mathbf{z}; \boldsymbol{\mu}_k, \boldsymbol{\Sigma}_k) \tag{1}$$

where $K$ is the total number of Gaussians in the mixture, and $k^{\text{th}}$ component is characterized by a Gaussian distribution with weight $\alpha_k$, mean $\boldsymbol{\mu}_k$ and covariance matrix $\boldsymbol{\Sigma}_k$.

Given $x$ is the data and $k$ is the (latent) index of the mixture density, GMM allows us to compute the conditional likelihood $p(x|k, \theta)$ as well as responsibility probability $p(k|x, \theta)$. In the Bayesian paradigm, one may refer $p(k|x, \theta)$ as the posterior density where the prior is $p(k) \equiv U(\{1, \ldots, K\})$.

This will further allow us to retrieve many important statistics and properties about the dataset, for example the segment membership of the data, the overall weights of the modes, we may even synthesizing meaningful "out-distribution" data using a convex combination of any two modes.

However, the conditional likelihood is hard to compute when dealing with high dimensional data, such as images, as data in its raw form do not form mixtures naturally.

At the same time, Generative Adversarial Network (GAN) gives us a way to compute the latent representation $z$ associated with the data $x$: GAN (Goodfellow et al. (2014)), introduces a 2-player non-cooperative game by a generator $G$ and a discriminator $D$. The generator produces samples from the random noise vector $\mathbf{z}$. The Discriminator differentiates between true samples and fake samples. The objective function of the game is given as follows:

$$\min_{G}\max_{D}V(D,G) = \mathbb{E}_{\boldsymbol{x}\sim p_{\text{data}}(\boldsymbol{x})}[\log D(\boldsymbol{x})] + \mathbb{E}_{\boldsymbol{z}\sim p_{\boldsymbol{z}}(\boldsymbol{z})}[\log(1 - D(G(\boldsymbol{z})))] \tag{2}$$

The problem of the original GAN methodology is that the association between the latent vector $\mathbf{z}$ from a low-dimension latent space to a data sample $x$ (in high-dimension data space) is one way, i.e., one must generate $\mathbf{z}$ before generating $x$, making the backward association from $x \rightarrow z$ infeasible. Since the generation of $\mathbf{z}$ is independent of $x$, any arbitrary distribution should theoretically achieve its take, making a uniform distribution or a standard Gaussian distribution a popular choice.

In our proposed work, we are to devise methods in which we are able to compute $p(z|k,\theta)$ and $p(z|x,\theta)$ to replace $p(x|k,\theta)$ and $p(k|x,\theta)$ respectively, through the use of GAN training. $p(z|x,\theta)$ is achieved by upgrading prior distribution $p(z)$ to a GMM instead of a simple Gaussian. At the same time, the responsibility probability $p(k|x,\theta)$ is learned through a classification network which is trained from end-to-end with the GAN. The learned classifier can subsequently be used to classify new data points or perform segmentation on the test set. We also observe through experiments that a smooth linear interpolation can be performed across multiple distribution modes to create the desired effects of "out-distribution" data.

Below we introduce some previous research that are related to our work.

## 1.1 Gaussian Mixture Model in GAN

Eghbal-zadeh & Widmer (2017) integrate a GMM into the GAN framework. Both of the means and covariance matrices are trainable through the generator loss. Besides, instead of applying the classic adversarial loss as in Equation 1, the authors proposed to use GMM likelihood.

Ben-Yosef & Weinshall (2018) also proposed a method named GM-GAN which used Gaussian Mixture to model the distribution over the latent space, in addition to its variant for the conditional generation. The supervised GM-GAN modifies the Discriminator, so that instead of a single scalar, it returns a vector $\boldsymbol{o} \in \mathbb{R}^{N}$. Each element of $\boldsymbol{o}$ represents the probability of the given sample being in each class. This will allow that images generated from the generator will be classified by the discriminator as a certain class.

In addition, in the absence of prior knowledge, the GMM is assumed to be uniform for both works above, i.e., $\forall k \in \{1,\ldots,K\}$ $\alpha_k = \frac{1}{K}$. On the contrary, our work makes no such assumption, the segmentation of data can be inferred from the trained model.

## 1.2 Linear Interpolation in GAN

Previously researchers have studied linear interpolation in the trained GAN models, where linear interpolations in the noise space lead to semantic interpolations in the generated images. Bojanowski et al. (2017) models the latent space as a unit sphere and learns the correspondence from the sphere space to the data space without the adversarial loss. Such a trained model is able to achieve smooth linear interpolation output between any two random vectors on the unit sphere.

Chen et al. (2016) proposed InfoGAN which learns to maximize the mutual information between a subset of latent variables and the observation, i.e. $I(c; G(\mathbf{z}, c))$, where $c$ is the class of the real sample. A learned InfoGAN model can disentangle discrete and continuous latent factors. Thus, linear interpolation can be performed using the continuous latent code, for example, from a thin digit to a wide digit; but not possible on the discrete codes, such as across categories of images.

In the next section, we explain in details how each part of the proposed algorithm works.

### 1.3 PAPER ORGANIZATION

In Section 2, we describe how each component of the proposed mechanism works. In Section 3, we demonstrate the performance of the proposed method by comparing it against several baseline models. Section 4 concludes the paper.

## 2 ARCHITECTURE

The proposed architecture consists of three networks. First, a classifier $C$ which outputs the possibility that a given image belonging to each Gaussian, and the possibility is used to construct a GMM distribution. Second, a Generator $G$ which produces synthetic samples from GMM random vectors. Third, a Discriminator $D$ which encodes samples to feature vectors and discriminates between real and synthetic samples. The overall architecture design is shown below in Figure 1. In the following sections, we explain the details of the three networks.

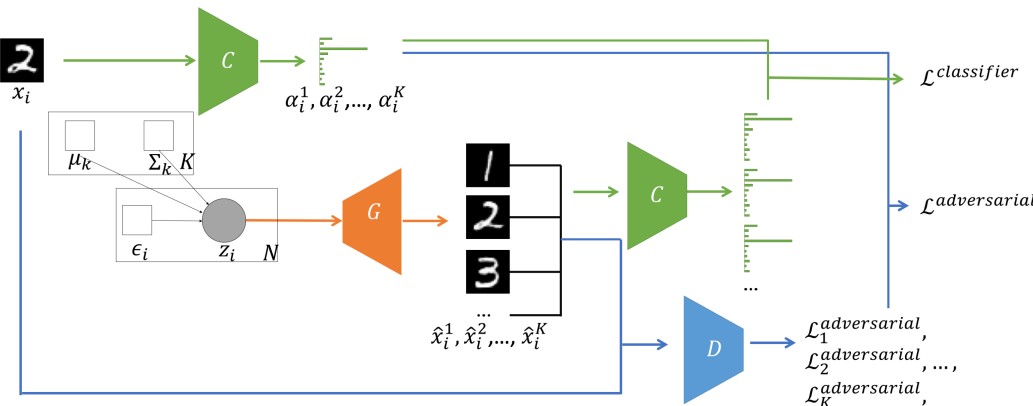

Figure 1: The overall architecture. The feed-forward logic of the Classifier $C$, the Generator $G$ and the Discriminator $D$ are marked with different colours.

### 2.1 THE CLASSIFIER

For an image $x_i$, the classifier $C$ outputs the probability $\boldsymbol{\alpha}_i = \{\alpha_i^1, \ldots, \alpha_i^k, \ldots, \alpha_i^K\}$ of the image belonging to each Gaussian indexed from 1 to $K$. These probability values are considered to be the mixture weights of the GMM which corresponds to the specific image, i.e, the density function of the GMM model which corresponds to image $x_i$ is $\sum_{k=1}^{K} \alpha_i^k \mathcal{N}(\boldsymbol{\mu}_k, \boldsymbol{\Sigma}_k)$. $\boldsymbol{\mu}_k$ and $\boldsymbol{\Sigma}_k$ for $k = \{1, 2, \ldots, K\}$ are trainable parameters. During test time, the learned Classifier can easily output $p(k|x, \theta)$ for any test image $x$.

The design of the classifier has two alternatives, the first is to share the feature encoding layers with the Discriminator. The shared feature encoding network encodes each image to a feature vector, and the Classifier will be a simple standard linear Soft-max classifier built on top of the features. The second is to build a stand-alone network which contains multiple CNN layers to classify images.

Details of both network designs can be found below in Table 1. In Table 1, we use acronyms for operations in the table: "Conv" is the convolution operation, of which the kernel and stride size are in the bracket; "Batch norm" is short for batch normalization; "Flatten" refers to the operation that flattens a tensor to 1D array. The same acronyms are also used in Table 2 and Table 3. $D_{\text{img}}$, $D_h$ and $D_z$ are related to the datasets we use, the exact value of each are reported in Section 3.

In Section 3.3, we report the results of both classifier designs in terms of generation performance and computation costs.

| Stage | Sub-stage | Name | Input Tensors | Output Tensors |
|---|---|---|---|---|
| $C$ | Encoding Network (if shared) | Conv (kernel=5, stride=2) + LeakyReLU | $D_{\text{img}} \times D_{\text{img}} \times D_h$ | $D_{\text{img}}/2 \times D_{\text{img}}/2 \times 64$ |
| | | Conv (kernel=4, stride=2) + Batch norm + LeakyReLU + Flatten | $D_{\text{img}}/2 \times D_{\text{img}}/2 \times 64$ | $D_{\text{img}}/4 \times D_{\text{img}}/4 \times 128$ |
| | Encoding Network (if not shared) | Conv (kernel=5, stride=1) + ReLU | $D_{\text{img}} \times D_{\text{img}} \times D_h$ | $D_{\text{img}} \times D_{\text{img}} \times 32$ |
| | | MaxPool (pool_size=2, stride=2) | $D_{\text{img}} \times D_{\text{img}} \times D_h$ | $D_{\text{img}}/2 \times D_{\text{img}}/2 \times 32$ |
| | | Conv (kernel=5, stride=2) + ReLU + Flatten | $D_{\text{img}}/2 \times D_{\text{img}}/2 \times 32$ | $D_{\text{img}}/4 \times D_{\text{img}}/4 \times 16$ |
| | | Linear + ReLU | $D_{\text{img}}/4 \times D_{\text{img}}/4 \times 16$ | 1024 |
| | Classification Network (if shared) | Linear | $D_{\text{img}}/4 \times D_{\text{img}}/4 \times 128$ | $K$ |
| | Classification network (if not shared) | Linear | 1024 | $K$ |

Table 1: The network structure of the Classifier $C$. Each column from left to right reports the network, sub-stage name, name of operations and shapes of input and output tensors.

Parameters of the classifier are optimized by both the adversarial loss and the mutual information loss, the details of both loss functions are discussed in Section 2.2, and how the update is performed is introduced in Section 2.4.

The Classifier is not required for generating new images after the model is fully trained. During testing, a random vector is sampled directly from the GMM model for the generation. The Classifier can then used to assign an unseen image to Gaussians and perform segmentation on the testing set.

## 2.2 THE GENERATOR

Given the classification result $\alpha_i$ for the input image $x_i$, ideally, multinomial sampling is performed to select one Gaussian out of $K$. The density function of this sampling is $\sum_{k=1}^{K} \alpha_i^k * N(\boldsymbol{\mu}_k, \boldsymbol{\Sigma}_k)$, and a random vector $z_i$ is sampled from the chosen Gaussian distribution to perform the image generation.

However, our design expects the training to be performed in an "end-to-end" fashion, and the sampling process would break the chain of the back-propagation from the adversarial loss to the classifier parameters. Therefore, we instead sample one random vector from each Gaussian represented as $\boldsymbol{z}_i = [\boldsymbol{z}_i^1, \ldots, \boldsymbol{z}_i^K]$. These vectors are used to generate $K$ synthetic images $[\hat{x}_i^1, \ldots, \hat{x}_i^K]$. $\boldsymbol{\alpha}_i$ is used to weigh the adversarial loss calculated from each pair of a generated sample $\hat{x}_i^k$ and the real image $x_i$.

In addition, the reparameterization trick is applied to the $\boldsymbol{z}_i$ sampling process so that the back-propagation can be used to update the parameters $\mu_k$ and $\Sigma_k$ of each Gaussian. Instead of sampling $\boldsymbol{z}_i^k \sim \mathcal{N}(\boldsymbol{\mu}_k, \boldsymbol{\Sigma}_k)$, we define $\boldsymbol{z}_i^k = \boldsymbol{A}_k \epsilon + \boldsymbol{\mu}_k \ \forall k \in [1, K]$, where $\epsilon$ is sampled as $\epsilon \sim \mathcal{N}(0, I)$, and $\boldsymbol{\Sigma}_k = \boldsymbol{A}_k \boldsymbol{A}_k^\top$.

The adversarial loss for the proposed framework is thus calculated over $K$ pairs of real and synthetic samples as:

$$\mathcal{L}^{\text{adversarial}} = \mathbb{E}_{x_i \in p_{data}} \left( \frac{1}{K} \sum_{k=1}^{K} \alpha_i^k \times (\log D(x_i) + \log(1 - D(\hat{x}_i^k))) \right) \tag{3}$$

In addition to the adversarial loss, we also encourage the classifier to restore the input classification output from the generated image. Therefore, we also employ mutual information loss between the classification result from the real and the generated image. Following InfoGAN by Chen et al. (2016) where the mutual information loss is formulated as $I(c, G(\boldsymbol{z}, c)) = \mathbb{E}_{c \sim P(c), x \sim G(\boldsymbol{z}, c)}[\log Q(c|x)] + H(c)$. where $c$ is the class of real sample. We define the mutual information loss in our work as:

$$\mathcal{L}^I = \mathbb{E}_{x_i \in p_{data}} \left( \frac{1}{K} \sum_{k=1}^{K} \alpha_i^k \times I(\boldsymbol{\alpha}_i, \hat{x}_i^k) \right) \tag{4}$$

The Generator structure used in our experiments is as below in Table 2. In the table, "LeakyReLU" which is short for "leaky rectified linear unit" and "Tanh" are activation functions.

| Stage | Name | Input Tensors | Output Tensors |
|---|---|---|---|
| $G$ | Linear + Batch norm + LeakyReLU + Reshape | $1 \times D_z$ | $D_{\text{img}}/4 \times D_{\text{img}}/4 \times 256$ |
| | Transposed Conv (kernel=5, stride=1) + Batch norm + LeakyReLU | $D_{\text{img}}/4 \times D_{\text{img}}/4 \times 256$ | $D_{\text{img}}/4 \times D_{\text{img}}/4 \times 128$ |
| | Transposed Conv (kernel=5, stride=2) + Batch norm + LeakyReLU | $D_{\text{img}}/4 \times D_{\text{img}}/4 \times 128$ | $D_{\text{img}}/2 \times D_{\text{img}}/2 \times 64$ |
| | Transposed Conv (kernel=5, stride=2) + Tanh | $D_{\text{img}}/2 \times D_{\text{img}}/2 \times 64$ | $D_{\text{img}} \times D_{\text{img}} \times D_h$ |

Table 2: The network structure of the Generator $G$.

## 2.3 THE DISCRIMINATOR

As we mentioned in Section 2.1, the design of the Discriminator has two options: whether or not the image encoding layers are shared with the classifier. The image encoding network is followed by a standard linear logistic regression to identify the given image to be real or fake. Details of the network are shown below in Table 3.

| Stage | Sub-stage | Name | Input Tensors | Output Tensors |
|---|---|---|---|---|
| $D$ | Encoding Network | Conv (kernel=5, stride=2) + LeakyReLU | $D_{\text{img}} \times D_{\text{img}} \times D_h$ | $D_{\text{img}}/2 \times D_{\text{img}}/2 \times 64$ |
| | | Conv (kernel=4, stride=2) + Batch norm + LeakyReLU + Flatten | $D_{\text{img}}/2 \times D_{\text{img}}/2 \times 64$ | $D_{\text{img}}/4 \times D_{\text{img}}/4 \times 128$ |
| | Discriminator Network | Linear | $D_{\text{img}}/4 \times D_{\text{img}}/4 \times 128$ | 1 |

Table 3: The network structure of the Discriminator $D$.

## 2.4 TRAINING PARAMETERS FOR THE PRIOR DISTRIBUTION

In our setting, the prior distribution is a GMM model, the two trainable variables are means for $K$ Gaussians $\boldsymbol{\mu} \in \mathbb{R}^{K \times D_z}$ and standard deviations for $K$ Gaussians $\mathbf{A} \in \mathbb{R}^{K \times D_z \times D_z}$. Both variables are updated by the adversarial loss in addition to the mutual information loss as the training is performed "end-to-end".

Below we give the pseudocode about how the updates are performed on each network in one iteration.

---
**Algorithm 1** Training the proposed model for 1 iteration

---
**Require:** $X = [x_1, x_2, \ldots, x_M]$ - $M$ training images in one batch
1: **for** $i = 1 \ldots M$ **do**
2:     Classify $x_i$ into $\alpha_i = [\alpha_i^0, \alpha_i^1, \ldots, \alpha_i^k, \ldots, \alpha_i^K]$
3:     **for** $k = 1 \ldots K$ **do**
4:         $\epsilon \sim \mathcal{N}(0, I)$
5:         $\boldsymbol{z}_k = \boldsymbol{A}_k \epsilon + \boldsymbol{\mu}_k$
6:         $\hat{x}_i^k \leftarrow G(\boldsymbol{z}_k)$
7:         Classify $\hat{x}_k$ into $\hat{\alpha}_i = [\alpha_i^0, \alpha_i^1, \ldots, \alpha_i^k, \ldots, \alpha_i^K]$
8:     **end for**
9:     Calculate $\mathcal{L}^{\text{adversarial}}$ from $x_i$ and $[\hat{x}_i^1 \ldots \hat{x}_i^K]$ as in Equation 3
10:     Calculate $\mathcal{L}^I$ from $\alpha_i$ and $\hat{\alpha}_i$ as in Equation 4
11:     Update the Discriminator with $\mathcal{L}^{\text{adversarial}}$
12:     Update both the Generator and the Classifier with $\mathcal{L}^{\text{adversarial}}$
13:     Update the Classifier with $\mathcal{L}^I$
14: **end for**

---

## 3 EXPERIMENTS

In this section, we evaluate the performance of the proposed method by comparing it with several baselines.

## 3.1 EXPERIMENT SETUP

The datasets we use are the MNIST (LeCun et al. (2010)), Fashion-MNIST (Xiao et al. (2017)) and Oxford-102 Flower (Krizhevsky (2009)) datasets. The details are listed below in Table 4. In

particular, we only select a subset of Oxford-102 to perform the training, which is the images which belong to the first 10 classes. For experiments performed on each dataset, we used different hyper-parameters, the details are listed in Table 5.

| Dataset | Number of Classes | Data Dimension | Train Samples | Validation Samples | Test Samples |
|---|---|---|---|---|---|
| MNIST | 10 | $28 \times 28 \times 1$ | 60,000 | - | 10,000 |
| Fashion-MNIST | 10 | $28 \times 28 \times 1$ | 60,000 | - | 10,000 |
| Oxford-102 Flower | 102 | $64 \times 64 \times 3$ | 1,020 | 1,020 | 6,149 |

Table 4: Statistics of the different datasets used in the empirical evaluation.

| Dataset | Number of Epochs | Learning Rate $\gamma$ | $D_{\text{img}}$ | $D_{\text{h}}$ |
|---|---|---|---|---|
| MNIST | 200 | 0.0002 | 28 | 1 |
| Fashion-MNIST | 200 | 0.0002 | 28 | 1 |
| Oxford-102 | 10,000 | 0.0002 | 64 | 3 |

Table 5: Hyper-parameters for the training performed on each dataset.

## 3.2 Linear Interpolation across Gaussian

Below in Figure 2, we show samples generated by the proposed model trained on several datasets in the completely unsupervised manner. We set the number of Gaussians equal to the total number of classes of the dataset in all experiments. When we are performing the linear interpolation as in the right panels, the random vector $z$ for each image generation is calculated as $z = A_k \epsilon + \mu_k$ $\forall k \in \{1, \ldots, K\}$, where $\epsilon$ is sampled as $\epsilon \sim \mathcal{N}(0, I)$. $\epsilon$ is kept the same for all images for each dataset.

We can draw two conclusions from the results in Figure 2. First, a fully trained proposed method can learn to "allocate" each class of image to a Gaussian. Second, the trained model can be used to perform smooth linear interpolation between Gaussians and even among more than two Gaussians. In Figure 3, we demonstrate the linear interpolation performed over 3 categories. The proportion of Gaussians of the synthetic images can be set manually.

## 3.3 Image Generation Quality

The generation performance is measured with two commonly used metrics: Inception score (Salimans et al. (2016)) and Frchet Inception Distance (FID) score (Heusel et al. (2017)).

*Inception score* is calculated as $I = \exp(\mathbb{E}_x D_{\text{KL}}(p(y|x)||p(y)))$, and a higher value generally indicates a better performance.

where $x$ is a generated image and $y$ is the label predicted by the Inception model (Szegedy et al. (2015)).

*FID score* is another metrics that measures the image generation quality. A lower value shows a better image quality and diversity. It calculates the difference between real images $x$ and generated images $g$ as $\text{FID}(x, g) = ||\mu_x - \mu_g||_2^2 + \text{Tr}(\Sigma_x + \Sigma_g - 2(\Sigma_x \Sigma_g)^{\frac{1}{2}})$.

Note that limitations of both Inception and FID score have been pointed out in several previous literature (Barratt & Sharma (2018), Lucic et al. (2018)), and there is currently no "perfect" metrics at this moment. These two metrics are used as an indication rather than a hard measure.

This evaluation is performed on the Oxford-102 dataset. As the Inception score is suggested to be evaluated on a large enough number of samples, we generate 5K synthetic samples to calculate both values. Below in Figure 4, we show the plot of Inception scores and FID scores calculated over the training epochs. In Table 6 we report the number of trainable parameters, the best Inception and FID score.

While the four algorithms use the same network settings for the Generator and Discriminator, it is clear to see that the proposed method, whether or not the encoding layers are shared between the

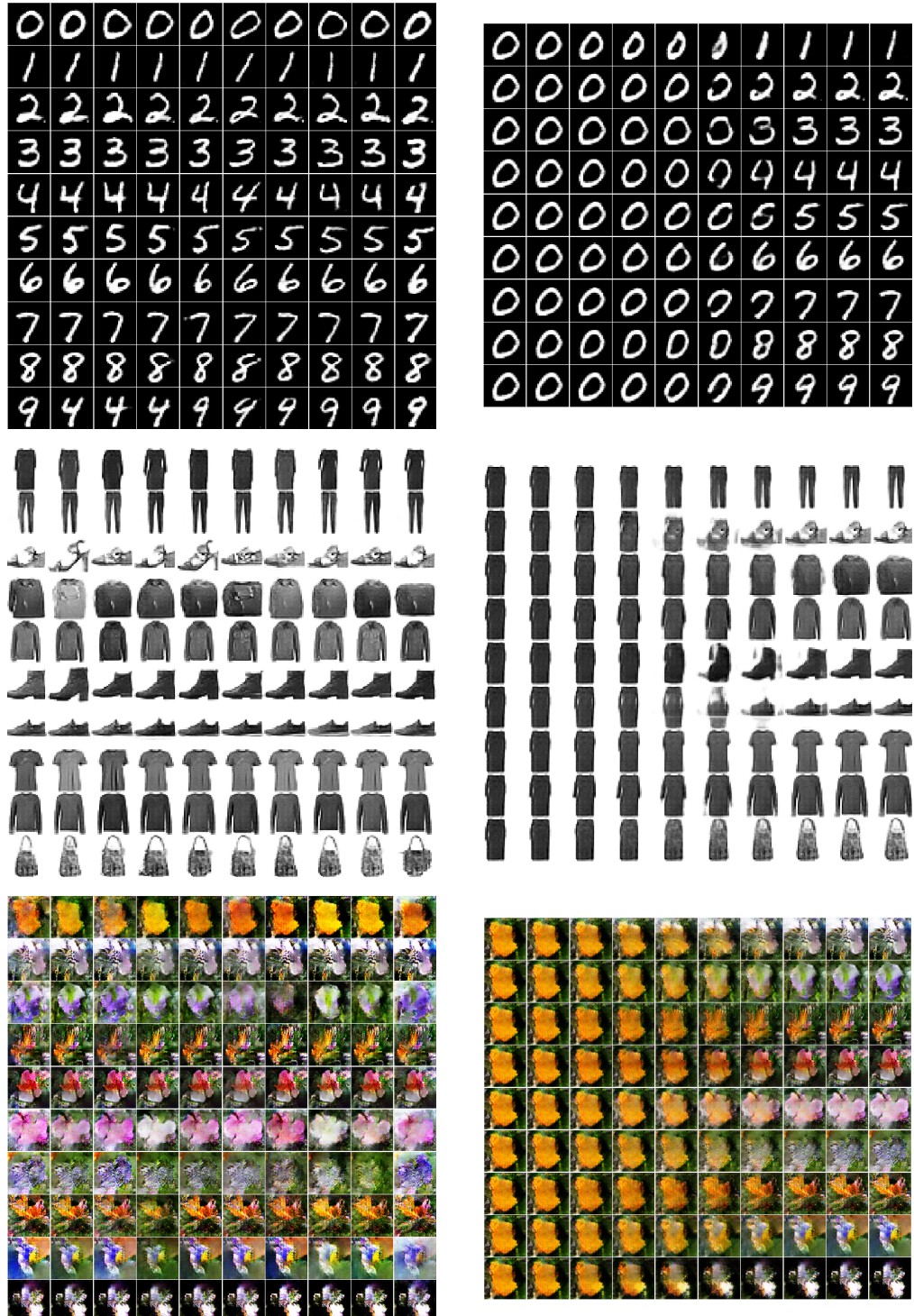

Figure 2: Samples generated by the proposed models trained on the MNIST (top panels), Fashion-MNIST (middle panels) and CIFAR-10 (bottom panels) datasets. In the left panels, the Gaussian mixture contains $K = 10$ Gaussians, and each row contains images sampled from a different Gaussian. In the right panels, each row shows the linear interpolation result from one Gaussian to another. Note that the index of Gaussian does not necessarily correspond to the actual digit, generated images are reordered as the order of the digit only for demonstration purpose.

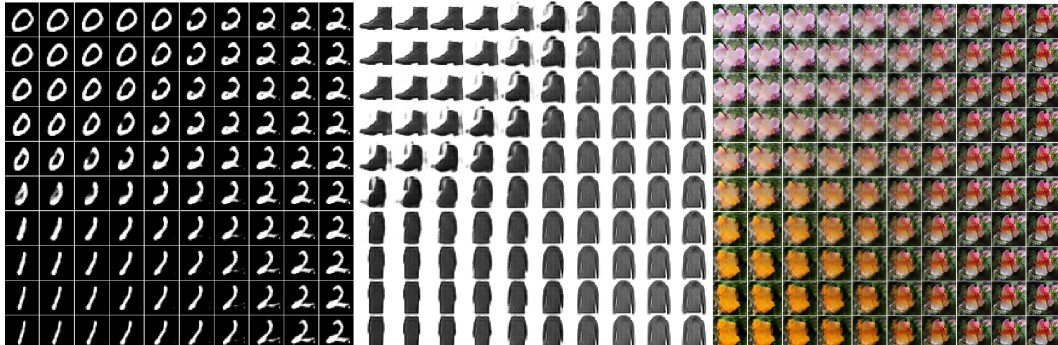

Figure 3: Linear interpolation over 3 Gaussians on the MNIST, Fashion-MNIST and Oxford-102 dataset. Each row and each column contains linear interpolation performed between 2 Gaussians. Images along the diagonal shows the interpolation across 3 Gaussians.

classifier and the Discriminator, is able to out-perform previous baseline models in terms of the image generation quality. The shared feature encoding layers would further improve the performance and reduce the size of the network.

| | number of parameters | Inception Score ↑ | FID score ↓ |
|---|---|---|---|
| Proposed (encoding not shared) | $13,005,411$ | $2.9664 \pm 0.2188$ | $231.0577 \pm 7.5371$ |
| **Proposed (encoding shared)** | $8,794,835$ | $\mathbf{3.1368 \pm 0.1596}$ | $\mathbf{205.9776 \pm 7.8587}$ |
| GM-GAN | $8,467,145$ | $2.6770 \pm 0.1079$ | $239.3936 \pm 6.7672$ |
| Vanilla GAN | $8,366,145$ | $2.4882 \pm 0.1065$ | $247.0610 \pm 7.2361$ |

Table 6: Number of parameters, Inception scores and FID scores of the proposed method and the baselines.

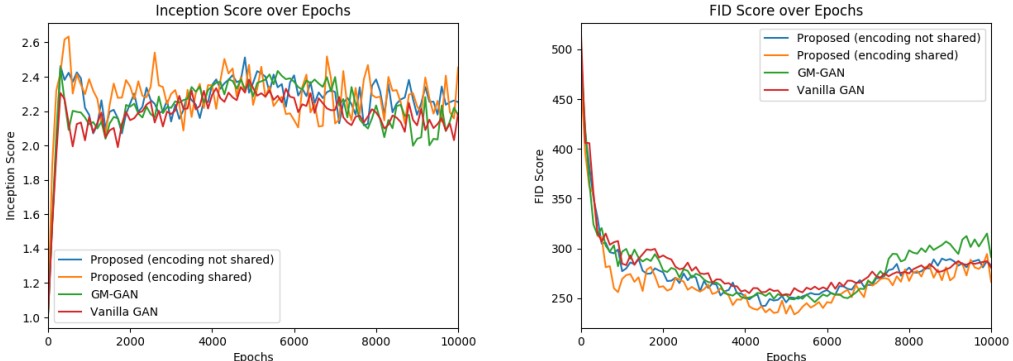

Figure 4: Inception score and FID score over training epochs.

## 4 CONCLUSION

In this paper we propose a novel framework which we incorporate the GMM to model the latent prior distribution. We also use a Classifier that learns to categorize an image to Gaussians. The Classifier is trained with the GAN model in an "end-to-end" fashion. We demonstrate through experiments that the proposed method surpass previous baselines in terms of the image generation performance with only minor growth on the size of the network. A trained model is also able to perform smooth linear interpolation across Gaussians, i.e. generate images with mixed styles from multiple categories.

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
