# OpenReview forum: "GAN-based Gaussian Mixture Model Responsibility Learning"
_ICLR.cc/2020/Conference — Reject_

### Official Review · AnonReviewer2 · 2019-10-24
**Official Blind Review #2**

**Rating:** 1

**Review:**

This paper proposes a modification of GANs where the latent space follows a distribution modelled by a Gaussian Mixture Model. While the idea of using GMMs in GANs is not novel, the main contribution of the paper is to add a classification models that enables posterior inference. The whole model is trained jointly end-to-end using both an adversarial loss and a mutual information loss. The procedure is then tested on MNIST, Fashion-MNIST and a subset of Oxford-102 Flower.

While the problem of posterior inference is interesting, the novelty of the paper is quite limited. The overall structure is clear, although writing can be improved.

However, a part from the form, I have other concerns about the paper:
- The main purpose of the paper is to tackle large image datasets that are hard to address by classical GMMs. This being said, the used datasets are composed of small and modal enough images that it seems hard to validate the claim of the paper using only these data. It seems to me that for the claim of the paper to be verified, larger scale/more complex datasets are needed.
- In all experiments, the authors suppose they have access to the number of classes/modes in the data, which is a huge assumption. It would be interesting to see if it would be possible to automatically accurately select the number of modes, e.g. on a held out validation set.
- One problem of GANs that the authors do not seem to consider is mode collapse. It would be interesting to do experiments with unbalanced datasets (e.g. MNIST 1 vs all) to see if the proposed architecture will model the data correctly.
- I am confused about the use of Mutual Information loss. The author claim that they would like to enforce each of the generated images to be from the same class as the input image. This would make sense if the authors used the multinomial sampling for the latent variable generation. However, the K generated samples are from K different Gaussians. It seems unreasonable to require of the classifier to render the same result.
- On the same note, in order to both use the classical multinomial sampling in GMMs and not break the backpropagation, have the authors considered updating the classifier and the GAN in separately in expectation-maximization fashion?
- Finally, I don't see the point of weighting the adversarial loss by the weights of Gaussians. All the generated images are fake and should be equally detected as such.
Although the idea is interesting, I think the paper, at its current status, not ready for publication.

Minor:
- P. 4: Generator: The sampling density from the multinomial distribution seems incorrect. However, as the authors skip the sampling step to be able to back propagate through the model, this is not significant.
- P. 1: The paragraph before the last: may even synthesizing -> synthesize
- P. 3: Architecture: Possibility -> Probability
- P. 7: Figure title: CIFR10 -> Oxford-102
- Algorithm notation:
    *Indexes for alpha and alpha_hat from 1 and not 0 to be consistent with the rest of the text
    *Add hats to the entries of alpha_hat
    *  LI from alpha_i and alpha_hat_i as in Equation 4 -> maybe change alpha_hat by x_hat to be consistent with the notation of eq.4 (although the meaning is clear here).


**Experience Assessment:**

I have read many papers in this area.

**Review Assessment: Checking Correctness Of Derivations And Theory:**

N/A

**Review Assessment: Checking Correctness Of Experiments:**

I carefully checked the experiments.

**Review Assessment: Thoroughness In Paper Reading:**

I read the paper at least twice and used my best judgement in assessing the paper.

---

### Official Review · AnonReviewer1 · 2019-10-25
**Official Blind Review #1**

**Rating:** 3

**Review:**

This paper considers the Gaussian mixture model at the latent space to have a better GAN training result.  The proposed architecture consists of three networks, a classifier, a generator, and a discriminator. Every input image goes through the classifier and gets the softmax output. The softmax output is considered as the mixture weights of the GMM model and controls the loss function of the generator accordingly.

Although this paper has some positive sides, I recommend "weak reject" because of the following reasons.

1. This paper is hard to follow. Many concepts are not discussed enough. For instance, how to train mu_k and \Sigma_k, why should we use Eq. (3) as the loss of the generator, where do we use L^I in Eq.(4),...

2. The experiment section requires more works. It would be much better to do experiments with much higher dimensional data sets and compare with many other GAN algorithms e.g. InfoGAN.

3. The pseudo-code has many errors. For instance,  what is \alpha_i^0? It is not introduced. What is \hat{\alpha}?

**Experience Assessment:**

I do not know much about this area.

**Review Assessment: Checking Correctness Of Derivations And Theory:**

N/A

**Review Assessment: Checking Correctness Of Experiments:**

I did not assess the experiments.

**Review Assessment: Thoroughness In Paper Reading:**

I made a quick assessment of this paper.

---

### Official Review · AnonReviewer3 · 2019-10-28
**Official Blind Review #3**

**Rating:** 1

**Review:**

This paper proposes to use GMM as the latent prior distribution of GAN. The model adds the means, covariances and the discrete priors of the GMM as learnable parameters to the GAN, which are jointly optimized with other GAN parameters. The model also adds a discrete classifier to the training process. During training, the classifier predicts the probability of each image falling into each of the GMM clusters, and uses these probabilities to re-weight the GAN generated samples.

# motivation

The motivation of this work is not clear to me. Even with an isotropic Gaussian prior, a fully connected neural network is already sufficient to (approximately) simulate the GMM sampling. Thus, explicitly modeling GMM doesn't seem to be necessary, and could make the learning more difficult.

I also don't quite understand why the authors have to add a discrete classifier to the modeling. It appears that the discrete classifier is only used for controlling the relative weights of clusters in the GAN training. If that's the case, then what's truly needed is just the prior distribution of each cluster, which doesn't depend on the individual images. For concrete datasets, this prior is usually known. For example, in MNIST each cluster has an equal prior of 10%.

# experiments

The model is evaluated on MNIST and Oxford-102. I'd like to see it tested on more realistic and higher resolution images, and compared with state-of-the-art GAN models. Since the motivation of the modeling design is unclear, the bar on the empirical results should be much higher.


**Experience Assessment:**

I have read many papers in this area.

**Review Assessment: Checking Correctness Of Derivations And Theory:**

I assessed the sensibility of the derivations and theory.

**Review Assessment: Checking Correctness Of Experiments:**

I assessed the sensibility of the experiments.

**Review Assessment: Thoroughness In Paper Reading:**

I read the paper at least twice and used my best judgement in assessing the paper.

---

### Decision · Program_Chairs · 2019-12-19

**Decision:**

Reject

**Comment:**

This paper proposes to use GMM as the latent prior distribution of GAN. The reviewers unanimously agree that the paper is not well motivated, explanations are lacking and writing needs to be substantially improved.